# HIV-Induced Thymic Insufficiency and Aging-Related Immunosenescence on Immune Reconstitution in ART-Treated Patients

**DOI:** 10.3390/vaccines12060612

**Published:** 2024-06-04

**Authors:** Maria Carolina Santos Guedes, Wlisses Henrique Veloso Carvalho-Silva, José Leandro Andrade-Santos, Maria Carolina Accioly Brelaz-de-Castro, Fabrício Oliveira Souto, Lílian Maria Lapa Montenegro, Rafael Lima Guimarães

**Affiliations:** 1Department of Genetics, Federal University of Pernambuco—UFPE, Recife 50670-901, PE, Brazil; mariacarolina.guedes@ufpe.br (M.C.S.G.); rafael.lguimaraes@ufpe.br (R.L.G.); 2Keizo Asami Institute (iLIKA), Federal University of Pernambuco—UFPE, Recife 50670-901, PE, Brazil; wlisses.veloso@ufpe.br (W.H.V.C.-S.); jlandrades19@gmail.com (J.L.A.-S.); fabricio.souto@ufpe.br (F.O.S.); 3Aggeu Magalhães Institute—Oswaldo Cruz Fundation (IAM/FIOCRUZ), Recife 50740-465, PE, Brazil; carolina.brelaz@gmail.com; 4Vitória Academic Center (CAV), Federal University of Pernambuco—UFPE, Recife 55608-680, PE, Brazil; 5Agreste Academic Center (CAA), Federal University of Pernambuco—UFPE, Recife 55014-900, PE, Brazil

**Keywords:** immunological non-responders, antiretroviral therapy, CD4+ T cell reconstitution, thymic exhaustion

## Abstract

The mechanisms underlying unsatisfactory immune reconstitution in HIV-1 positive patients under ART have not been fully elucidated, even after years of investigation. Thus, this study aimed to assess the correlation between age and thymic production profile, and its influence on inadequate immunological recovery. Here, 44 ART-treated patients with undetectable plasma HIV-1 load (<40 copies/mL) were classified as 31 immunological responders (IR) and 13 immunological non-responders (INR), according to their CD4+ T-cell count after 18 months of ART. The thymic function was assessed by identifying recent thymic emigrants (RTEs) CD4+ T cells (CD4+/CD45RA+CD31+) in PBMCs using flow cytometry. Clinical data were also analyzed from medical records. The INR group showed a higher age at ART initiation (41 ± 3.0) compared to the IR (33.7 ± 2.1) group (*p* = 0.041). Evaluating RTE CD4+ T-cells, we observed a lower percentage in the INR group (19.5 ± 6.3) compared to the IR group (29.9 ± 11.5) (*p* = 0.012). There was a strong negative correlation between age at ART initiation and RTE CD4+ T-cells in INRs (r = −0.784, *p* = 0.004). Our study has highlighted the thymic insufficiency and aging-related immunosenescence with unsatisfactory immunological recovery during ART in HIV-1 positive patients.

## 1. Introduction

Acquired immunodeficiency syndrome (AIDS) is the final stage of the infection caused by human immunodeficiency virus type 1 (HIV-1), the agent responsible for compromising the immune system through the destruction of CD4+ T lymphocytes [1]. Therefore, antiretroviral therapy (ART) aims to inhibit virus replication to undetectable levels, making the individual non-transmissible and preventing disease progression and AIDS-related deaths [2]. Although significant progress has been evidenced in the treatment of HIV-1 positive patients, certain aspects remain partially elucidated, such as the insufficient immune reconstitution observed in patients undergoing ART [3,4]. 

Even though achieving complete virological suppression (<40 RNA copies/mL), 10–40% of ART-treated patients are unable to restore the lost CD4+ T lymphocytes over the infection’s course. Defined as immunological non-responders (INR), these individuals are more susceptible to developing neoplasms and opportunistic infections, as well as experiencing various immune dysregulations, including altered profile of cytokine secretion and regulatory components (T-reg and Th17), mitochondrial disruption, and immunosenescence [5,6,7]. 

While the mechanisms contributing to deficient immunological recovery remain an area of ongoing investigations, several factors have been implicated over time. These include male sex, reduced pre-ART CD4+ T lymphocytes count, advanced age, persistent viral replication, coinfections, and genetic variants [7,8,9]. Furthermore, it is suggested that immune reconstitution depends on a delicate equilibrium between thymic production and CD4+ T lymphocytes death. A decline in thymic activity and/or an increase in cellular depletion are critical in sustaining the state of immunodepression [9,10].

During ART, the increase in CD4+ T lymphocytes may occur in a progressive and gradual manner, with a rapid increase in the first months and becoming less progressive over the course of treatment. Several mechanisms are necessary for this cellular reestablishment, including production, activation, proliferation, and differentiation pathways of CD4+ T lymphocytes [9]. Additionally, some studies have shown that pyroptosis, a highly inflammatory type of cell death, is the main pathway of CD4+ T cells depletion in HIV-1 infection [11]. However, it is believed that thymic production is one of the primary mechanisms responsible for the reconstitution of CD4+ T-cells within the first two years of ART [9]. The evaluation of its activity often involves assessing the output of recent thymic emigrants (RTE). Studies have indicated that the INR group exhibits lower levels of production of these cells, demonstrating impaired thymopoiesis [12,13]. This mechanism can be influenced by various factors, including age [14]. Thus, this study aimed to investigate the correlation between age and thymic production profile, and its influence on inadequate immunological recovery.

## 2. Materials and Methods

### 2.1. Study Population

Here, 44 ART-treated HIV-1-positive patients (11 females and 33 males) were recruited between 2018 and 2020 at Professor Fernando Figueira Integral Medicine Institute (IMIP), Pernambuco state (Northeast Brazil). The general inclusion criteria were age ≥18 years old, being on ART for at least 18 months, demonstrating good adherence, and maintaining undetectable HIV-1 load (<40 copies/mL) at all clinical appointments after therapeutic success (complete viral suppression within six months [15]). In addition, the general exclusion criteria were history of injecting drug use, autoimmune diseases or any other immunological complications, cancer, and pregnancy. Clinical data were evaluated from medical records. All patients answered standard questionnaires and signed written informed consent forms. This study was approved by the Ethics Committee of Instituto de Medicina Integral Professor Fernando Figueira (protocol number: 3629-13).

### 2.2. Patient Classification

The ART-treated patients were classified regarding CD4+ T lymphocyte count changes. Patients who initiated ART with a CD4+ T-cell count ≥500 cells/µL or attained a CD4+ T-cell count exceeding this threshold after 18 months of ART were categorized as immunological responders (IR). Conversely, patients who started ART with a CD4+ T-cell count <500 cells/µL and continue to have CD4+ T-cell levels below this threshold even after 18 months of ART were classified as INR (adapted from [5]). Therefore, 13 ART-treated patients met the criteria for being defined as INR, while the remaining 31 HIV-1 positive patients under therapy were categorized as IR.

### 2.3. PBMC Isolation and Flow Cytometry Analysis

Blood samples were obtained from all participants using EDTA tubes (4 mL). Afterward, peripheral blood mononuclear cells (PBMCs) were separated via density gradient centrifugation with Ficoll–Paque Plus, followed by two washes using PBS 1x. The PBMCs were resuspended in FACS buffer consisting of 3% BSA, 0.01% sodium azide (NaN_3_), and PBS 1x. Cell viability, averaging over 95%, was assessed using the Trypan blue exclusion test at 0.4%. The PBMCs were labeled with a combination of fluorescent monoclonal antibodies including APC-CD4 (EDU-2), PercpCy5.5-CD45RA (HI100), and PE-CD31 (WM59) obtained from BD Biosciences. After incubating at room temperature for 20 min (protected from light), the cells were washed with FACS buffer and fixed in 1% PBS-formaldehyde. Flow cytometry analysis was performed using a FACSAria III cytometer (BD Biosciences, Franklin Lakes, NJ, USA), acquiring 50,000 events for lymphocytes and gating (Figure 1A) to detect RTE CD4+ T cells (CD4+/CD45RA+CD31+). The acquired data were analyzed using the FlowJo software, version 10.

### 2.4. Statistical Analysis

The normal distribution of the data was assessed using the Shapiro–Wilk test. Variables with a normal distribution have been presented as mean ± SD, and group comparisons were conducted using Student’s *t*-test. For variables that did not follow a normal distribution, median and interquartile ranges (IQR) were reported, and the Mann–Whitney test was employed for analysis. Correlation analyses were conducted using the Pearson correlation test. A significance level (α) of 0.05 was set for all tests, which were two-tailed. Statistical analyses were performed using GraphPad Prism, version 8.0.1.

## 3. Results

Analyses of clinical characteristics revealed a statistically significant difference in the mean age at ART initiation, being higher in the INR (41 ± 3.0) group when compared to the IR (33.7 ± 2.1) group (*p* = 0.041). Additionally, we also observed significant differences between INR and IR groups regarding pre- and post-treatment CD4+ and CD8+ T-cell count, as well as percentages (Appendix A). Other variables such as sex, body mass, time to ART starting post-diagnosis, pre-treatment PVL, and ART regimens were also analyzed; however, there were no significant difference between the groups.

Concerning flow cytometry analyses, the evaluation of thymic function demonstrated a lower percentage of RTE CD4+ T-cells in the INR group (19.5 ± 6.3) compared to the IR group (29.9 ± 11.5), showing statistical significance (*p* = 0.012, Figure 1B).

Correlation analyses between age and RTE CD4+ T-cells were performed in the INR and IR groups. It was possible to observe a strong negative correlation between age at ART start and RTEs levels in the INR group (r = −0.784, *p* = 0.004, Figure 1D). Regarding IR group, a weak negative correlation (r = −0.269, Figure 1C) was observed, but it was not statistically significant (*p* = 0.193).

## 4. Discussion

Since the development of the first antiretroviral drug, important advances have been made in HIV-1 treatment, improving the quality of life for many patients. Thus, the goal of ART is no longer just to control viral replication, but also to provide a favorable environment for immune reconstitution [16]. Nevertheless, a significant percentage of these patients continue with a low count of CD4+ T-cell, even after virological suppression, and remain susceptible to coinfections [7]. Diverse mechanisms are responsible for CD4+ T-cell restoration; among them, thymopoiesis is one of the most relevant, as it not only increases cell count but also partially restores the patient’s antigenic repertoire. This thymic contribution can vary according to age [9,17]

Thymopoiesis is most active in early life, declining gradually with age (approximately 1–3% per year) due to thymic involution, and becoming less functional in adulthood, mainly after 40 years of age [18,19,20]. Thus, the thymus production activity in adults is considered limited [17]. However, under conditions of the massive depletion of T-cells and severe immunodeficiency, such as HIV-1 infection, the thymus reactivates, releasing RTEs to replenish and maintain peripheral lymphocytes [17]. Studies, including some from our research group, have demonstrated thymic production to be crucial for immune reconstitution during ART, particularly in patients with lower CD4+ T-cell counts [12,13,21].

In this study, our findings evidence a reduced output of RTEs in the INR group compared to the IR group, suggesting insufficient thymic function. Moreover, it has been observed that this group starts ART at a more advanced age, corroborating with studies that associated older age to poor immunological recovery [18,22]. This could be a result of delayed diagnosis leading to prolonged untreated periods, elevating viral replication, cellular depletion, and thymic exhaustion. Consequently, it can impair immunological recovery by significantly reducing the generation of RTE CD4+ T cells [7,23]. Furthermore, our study has shown a strong negative correlation between patients’ age at ART initiation and RTEs output, suggesting compromised thymic function with advancing age in these ART-treated patients. These findings even further confirm the importance of early diagnosis and ART initiation, as recommended by the World Health Organization guidelines [2].

As a compensatory mechanism for extensive cell depletion, the production of thymocytes by bone marrow occurs. However, HIV-1 can infect and kill thymocytes at various maturation stages, disrupting thymopoiesis and altering thymic tissue structure. This occurs even after virological suppression, since residual HIV-1 replication in reservoirs still persists. These alterations are similar to the thymic atrophy caused by advancing age [17,24]. Additionally, the process of thymocyte infection can also lead to exacerbated immune activation, another risk factor for insufficient immune reconstitution. In this case, the increase in immune activation was also demonstrated in the INR group in other studies from our research group [13]. This activation profile has also been shown to be more pronounced in older patients [18,25].

Alongside the thymic insufficiency induced by HIV-1, natural immunosenescence could also impede satisfactory immune reconstitution. This is because the changes associated with immunosenescence, such as increased inflammation and decreased T lymphocytes functionality, contribute to immune system dysregulation [19,26]. Moreover, senescent cells secrete various mediators, including pro-inflammatory cytokines, which can elevate inflammation and cell death. Consequently, this process can influence incomplete immunological recovery [14,22,26].

In addition, HIV-1 infection also causes a series of changes in the immune system that persist even after treatment adherence. In this context, these changes in immune response are similar to alterations observed in the adaptive immune system during the aging process [27]. Therefore, individuals living with HIV-1, even under ART, may develop an immune response with characteristics of an aged immune system, meaning these patients may exhibit an immunosenescent profile. This phenomenon of immunosenescence may be involved in incomplete immune reconstitution, as T lymphocytes are the main cells whose functions are compromised with aging [28,29]. It is important to highlight that those changes associated with immunosenescence, such as thymic dysfunction, lymphocyte activation, cell death, and reduced potential for cellular regeneration, have been described as more common in INRs [30]. Moreover, studies have suggested that HIV-1 infection can cause recurrent damage to cellular DNA and mitochondrial DNA. This damage can result in mitochondrial dysfunction, leading to cellular exhaustion, senescence, and ultimately cell death [31].

## 5. Conclusions

Our study has highlighted the role of thymic function and aging in immune reconstitution among patients undergoing ART, particularly in those with unsatisfactory immunological recovery. The reduced output of RTEs observed in the INR group emphasizes the importance of early diagnosis and ART initiation, since the aging also affects the thymic function. Additionally, our findings accentuate the impact of HIV-1-induced thymic insufficiency and aging-related immunosenescence on immune reconstitution.

## Figures and Tables

**Figure 1 vaccines-12-00612-f001:**
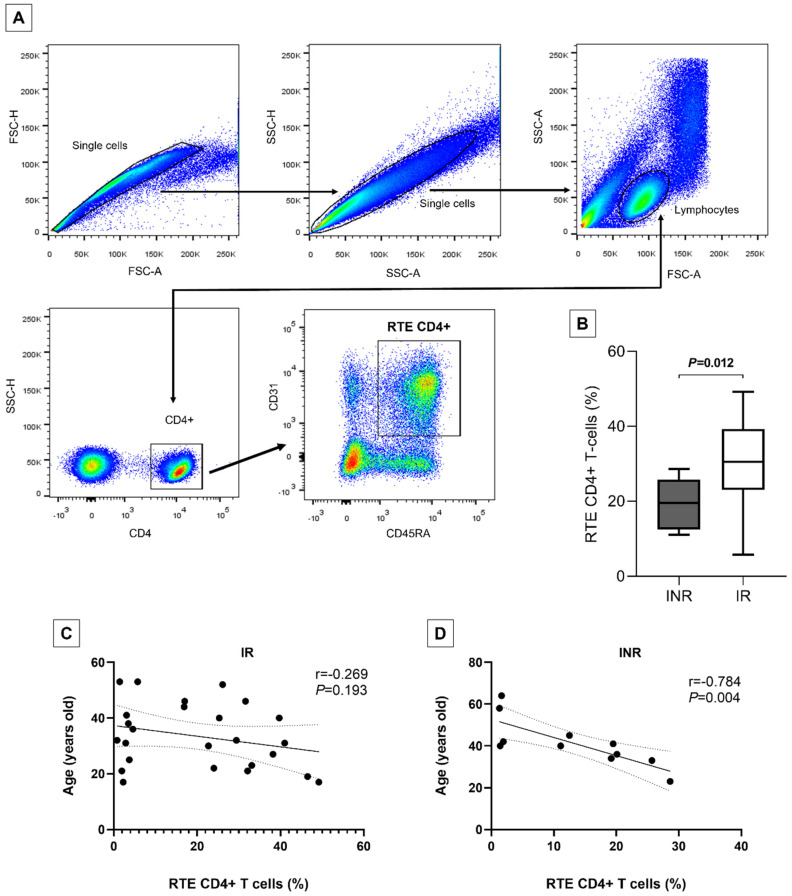
(**A**) The gating strategy was applied to identify RTE CD4+ T cells (CD4+/CD45RA+CD31+). (**B**) The percentage of RTE CD4+ T cells in the INR and IR groups is presented, including means, standard deviation, and *p*-value obtained from the t-test. Correlation between RTE CD4+ T cells and age (years old) at ART start date is shown for the IR (**C**) and INR (**D**) groups, with coefficient r and *p*-values according to Pearson’s correlation test. INR-immunological non-responders. IR-immunological responders. RTE-recent thymic emigrants.

## Data Availability

The data that support the findings of this study are available from the corresponding author upon reasonable request.

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
