# Peer review of "HIV-Induced Thymic Insufficiency and Aging-Related Immunosenescence on Immune Reconstitution in ART-Treated Patients"

_vaccines, 2024, doi:10.3390/vaccines12060612_

Round 1

Reviewer 1 Report

Comments and Suggestions for Authors

Manuscript ID:  vaccines-2935700

Title: HIV-induced thymic insufficiency and aging-related immunosenescence on immune reconstitution in ART-treated patients

Authors:  dos Santos Guedes et al.

In this manuscript, dos Santos Guedes et al. presents an evaluation of the correlation between age and the profile of thymic function and how influences immune reconstitution in HIV infection among patients of a clinic in Brazil between 2018 and 2020.

Overall, this is an interesting manuscript and provides scientific information to the body of knowledge. The Introduction contains enough information to provide some context. The methods used seem well considered. The results are presented thoroughly in the manuscript. The interpretation of the results is adequate and consistent with the reviewed findings. The Statements on Ethics, Data Availability and Conflicts of Interest are present and adequate.

However, the Reviewer would like to see the following issues addressed by the Authors:

Specific comments and suggestions:

METHODS

1.     There should be a mention of informed consent by patients in the Methods and information of approval from IRB or equivalent (protocol number, etc).

 RESULTS

1.     The Reviewer understands there is a statistical difference between the two groups, however when looking at the individual data points one can see that in the IR group there are individuals with ages 50 and late fifties. This would make one think that age is not the only factor affecting poor immune system reconstitution. The same way there are among the INR individuals in their 20s and 30s who should have better response if reconstitution was based only on age. Could you comment on which other factors could be at play here.

2.     Since the Authors indicated some individuals had less than 500 CD4s at start but passed that threshold by 18mo, one would like to see how the thymic function at start was compared to 18mo of ART. Did the Authors take a baseline sample at enrollment to measure thymic function at start to determine if there was change after 18mo of ART.

 DISCUSSION

1.   LINE 141: "Nevertheless, a significant of these patients" seems to be missing a word.

2.    LINE 150: Could you explain what you meant by "nearly unnecessary".

 CONCLUSION

1.     In hindsight one could find a statistically significant relationship between age and immune reconstitution in HIV infection. Please comment and elaborate on how this information could be used to identify among the newly diagnosed those that are at risk of diminished  immune reconstitution, despite successful virological suppression with ART, and how this information could be used to prevent poor outcomes in these patients.

 REFERENCES

There are multiple repeated references. This must be corrected.

1.    Citation #8 is repeated from Citation #3

2.    Citation #10 is repeated from Citation #5

3.    Citation #13 is repeated from Citation # 9

4.    Citation #21 repeated from Citation #5 and Citation #10

5.    Citation #26 repeated from Citation #22

6.    Citation #32 repeated from Citation #7 and Citation #15

Author Response

Dear Reviewer,

We appreciate all your suggestions, and our response to each one is attached.

Best regards,

Reviewer 2 Report

Comments and Suggestions for Authors

In the manuscript by Dos Santos Guedes and colleagues, the authors investigated thymic insufficiency in HIV-1 patients on anti-retroviral therapy for at least 18 months and had viral loads of <40 copies/mL.  The patients were divided into two groups; the immunological responders (IR; 31 patients) and immunological non-responders (INR; 13 patients), based on their CD4+ T cell count while on ART. The authors analyzed the levels of recent thymic emigrants (RTE) (CD4+CD3+ CD31+). The salient finding of this study was a strong negative correlation between the age at which ART is initiated and RTE CD4+ T cells in the INR population. Overall, the manuscript is well-written and would be of interest to the researchers in the HIV-1 field. 

1. An important control is missing. It would interest the readers if the authors examined the same cell types from age-matched healthy individuals (i.e., HIV-1[-]).  This comparison would give the readers an understanding of the normal rates of RTE cells entering the circulation.

2. “HIV” should be changed throughout the manuscript to “HIV-1.”

3. Line 182: In the sentence, “Consequently, the is process can influence on incomplete” should be changed to “Consequently, the is process can influence incomplete…”

4. Another control that would interest readers is the CD8+ RTE. This cell type is generally not infected by HIV-1 and expresses high levels of CD31+ on its surface.  

Comments on the Quality of English Language

No comments, a few typos.

Author Response

(The authors gave the same response as above.)
